# Preparation of PLGA-Coated Porous Silica Nanofibers for Drug Release

**DOI:** 10.3390/pharmaceutics14122660

**Published:** 2022-11-30

**Authors:** Meina Zhang, Jasmine Lidder, Mounib Bahri, Haifei Zhang

**Affiliations:** 1Department of Chemistry, University of Liverpool, Crown Street, Liverpool L69 7ZD, UK; 2Albert Crewe Centre for Electron Microscopy, University of Liverpool, Liverpool L69 3GL, UK

**Keywords:** organic nanofibers, silica nanofibers, self-assembly, sol–gel, freeze drying, drug release, poly(lactide-co-glycolide)

## Abstract

Fibrous materials have unique applications in drug release and biomedical fields. This study reports on the preparation of porous silica nanofibers, using organic nanofibers as templates, and their use for drug release. Different from the commonly used electrospinning method, the organic nanofibers are produced via a self-assembly approach between melamine and benzene-1,3,5-tricarboxylic acid. Silica is then coated on the organic nanofibers via homogenization in a silica sol, a freeze-drying process, and then a sol–gel process. In order to regulate the surface area and mesopore volume of silica nanofibers, cetyltrimethyl ammonium bromide at different concentrations is used as template in the sol–gel process. With the removal of organic nanofibers and the surfactant by calcination, porous silica nanofibers are generated and then assessed as a scaffold for controlled drug release with ketoprofen as a model drug. Poly (D, L-lactide-co-glycolide) is coated on the silica nanofibers to achieve slow burst release and prolonged cumulative release of 25 days. This study demonstrates an effective method of preparing hollow silica nanofibers and the use of such nanofibers for long-term release with high drug loading.

## 1. Introduction

Over the years, researchers have been exploring how to achieve a safe and desired drug delivery system. A range of natural and synthetic materials have been widely employed for drug delivery systems, such as biodegradable poly (D, L-lactide-co-glycolide) (PLGA) [1], biocompatible chitosan [2], metal organic frameworks (MOFs) [3], hydrogen-bonding organic materials [4] and ceramics [5]. These materials are produced for a variety of structures, including nanoparticles [6], nanofibers [7], and liposomes [8], as well as hydrogels [9].

Among the various materials investigated, silica is regarded as “Generally Recognized As Safe (GRAS)” by the Food and Drug Administration (FDA) of the USA. Porous silica materials have therefore been widely used for drug delivery and biomedical applications, due to the further capabilities of adjusting their pore size/shape and pore volume and modifying their surface functionality [10,11]. Various types of silica materials have been employed for drug delivery, e.g., sol–gel based silica [11], silica aerogels [12] and natural biosilica [13]. Among the different forms of silica materials, mesoporous silica nanoparticles have been widely investigated [6,10,14,15]. Silica nanoparticles can be further modified with surface coating or gates at the pore entrance in order to achieve a regulated or stimuli-triggered release [10,14]. For example, poly (acrylic acid) grafted mesoporous silica nanoparticles were prepared to load doxorubicin hydrochloride for cancer therapy [16]. There are extensive investigations where porous silica nanoparticles are embedded in microspheres, nanofibers, hydrogels and other formulations to maintain drug activity (particularly biopharmaceutics), improve biocompatibility/bioadhesion, and enhance controlled release profiles [17,18]. Silica-containing and silica nanofibers have also been used, due to the high surface area and highly accessible porosity in nanofibrous materials. For example, dendritic silica nanomaterials with a fibrous structure showed a high DNA adsorption capacity [19]. Luminescent porous silica nanofibers were produced by electrospinning and were utilized as a drug delivery host carrier [20]. There are additional advantages with the use of silica nanofibers. Silica nanofibers can be processed to generate 3D network structures, mimicking natural extracellular matrix (ECM) structure [21], which can be particularly useful when acting as a scaffold and drug reservoir for cell therapy and tissue engineering applications [21,22,23]. Electrospinning is a versatile method for preparing nanofibrous materials [24,25]. This has been used to produce polymer and composite nanofibers for biomedical and pharmaceutical applications [22,23]. Silica nanofibers cannot be directly prepared by electrospinning. However, they can be produced by generating polymer/silica composite fibers with subsequent calcination, or using the pre-prepared polymer nanofibers as templates [20,26,27]. Although the electrospinning set-up is straightforward and the approach is versatile, fine tuning of the preparation conditions is required to produce the desired nanofibers. Furthermore, the slow injection rate during electrospinning is a bottleneck step in producing a large quantity of nanofibers [24,25].

Self-assembly has been widely recognized as a versatile approach to fabricate functional and porous materials [28]. The assembled aggregates are generally formed by non-covalent bonds, such as hydrogen bonds, metal coordination, or other intermolecular interactions [29]. The physicochemical properties of self-assembled aggregates are quite different from the individual components, which allows these materials to be applied in a variety of fields [30]. Hydrogen-bonded organic frameworks are types of self-assembled materials which are formed via hydrogen-bonding interactions of organic building blocks [31]. Similarly, supramolecular organic frameworks are also generally synthesized by a self-assembling approach with permanent intrinsic porosity and chemical stability [32]. With the merits of mild synthesis conditions, good solution processability, and easy regeneration, self-assembly has been a powerful method to generate functional porous materials, nanofibers, and nanobiomaterials [27,33,34,35].

Motivated by the formation of nanofibers under mild conditions via a self-assembly process [33,34,35], often simply by mixing a solution or solution evaporation, we have investigated whether or not the self-assembly approach can be employed as an alternative to the traditional electrospinning method to fabricate nanofibers. Although a self-assembly method is unlikely to be as versatile as the electrospinning method, it may have advantages such as easy scale-up and fine control of fiber diameters. Hydrogen-bonding interaction is highly important in self-assembly, and the use of functional rigid molecules such as melamine and aromatic acids can facilitate the formation of crystalline nanostructures [29,32,36].

Herein, we report on the preparation of organic nanofibers, which are further used as a template to produce silica nanofibers for tuneable drug release. Melamine and benzene-1,3,5-tricarboxylic acid (BTC) were chosen as the building blocks to construct the melamine-BTC (M-BTC) nanofibers. Silica was then coated on the M-BTC nanofibers via homogenization, freeze-drying, and a sol–gel process. Subsequently, the M-BTC nanofibers were removed via calcination. During the sol–gel process, cetyltrimethylammonium bromide (CTAB) micelles served as a template to enhance the surface area and pore structure of the silica. This facilitated the uploading and release of a model drug, ketoprofen (KPF). To further tune the release behavior, PLGA was coated on the silica nanofibers to act as a release control barrier. This study developed an easy scale-up method for the preparation of organic and silica nanofibers. High drug loading into the silica nanofibers was achieved, which could act as a drug reservoir. A prolonged drug release profile with reduced burst release was then demonstrated.

## 2. Materials and Methods

### 2.1. Chemicals and Materials

Melamine (99%) was purchased from Alfa Aesar. Benzene-1,3,5-tricarboxylic acid (95%, BTC), tetraethylorthosilicate (98%, TEOS), oxalic acid (98%), as well as cetyltrimethylammonium (≥98%, CTAB) were bought from Sigma-Aldrich. Poly (D, L-lactide-co-glycolide) (PLGA) RG 503 (M.W. 24000) was purchased from Boehringer Ingelheim. *Tert*-butanol was provided by Merck. Acetone (≥99.8%,) and ethanol were supplied by Fisher Scientific. Ketoprofen (KPF) was purchased from Fluorochem Ltd. All chemicals were analytical grade and used without further treatment.

### 2.2. Self-Assembly of M-BTC Nanofibers

BTC (0.67 g) was dissolved in ethanol (50 mL) at 60 °C. Melamine (0.42 g) was added into deionized water (50 mL) with continuous stirring at 60 °C, until a clear solution was formed. These two solutions were combined together, and then 100 mL deionized water was added. After that, the mixture was stirred overnight at 90 °C and was cooled down to room temperature and then filtered out. Finally, the solid was frozen in liquid nitrogen and then freeze dried for 48 h using a freeze dryer (CoolSafe, Jencons-VWR).

### 2.3. Preparation of Silica Nanofibers

TEOS (4.0 g) was added into a water/*tert*-butanol mixture (volume ratios of water: *tert*-butanol 3:1, 100 mL). Oxalic acid (0.02 g, as a catalyst) was then added with continuous stirring for 4 h at room temperature to generate an homogeneous silica sol (4 *w*/*v*%) [37]. A different amount of CTAB was added into 8 mL silica sol (4 *w*/*v*%) and the mixture was stirred for 3 h at room temperature. The silica sol concentration was also varied (0.8 *w*/*v*%) to change the coating condition. M-BTC nanofibers (25 mg) were added into that solution. The mixture was homogenized for 8 min at 12,000 rpm (Power Gen 1000, Fisher Brand). After that, the mixture was frozen in liquid nitrogen and then freeze dried for 48 h. The freeze-dried materials were further treated in an oven for 24 h at 100 °C to complete the sol–gel process. The calcination process was carried out in a furnace (Carbolite, CWF1200) with the programmed procedure: heating at 2 °C/min to 600 °C, holding at 600 °C for 300 min, then cooling down at a rate 5 °C/min.

### 2.4. Drug Loading on Silica Nanofibers

KPF was dissolved in ethanol at the concentrations of 80 mg/mL and 20 mg/mL, respectively. Silica nanofibers (50 mg) were added in a 5 mL KPF solution at room temperature overnight (about 15 h). The silica nanofibers were then collected by filtration and placed in a vacuum oven at room temperature to remove all the solvent. The drug loading was calculated based on thermogravimetric analysis (TGA).

### 2.5. PLGA Coating on Silica Nanofibers

PLGA was dissolved in acetone at different concentrations (50 mg/mL, 100 mg/mL, and 150 mg/mL). KPF-loaded silica nanofibers were soaked in the PLGA solutions for different times (1 min, 3 min, and 5 min) and then taken out to be air dried in a fume cupboard. KPF was also dissolved in some PLGA solutions in order to prevent or reduce the leakage of KPF from silica nanofibers during the soaking process. KPF in PLGA coating could also provide a different release profile compared to the release from the silica nanofibers.

### 2.6. In Vitro Drug Release

20 mL phosphate buffer solution (PBS, pH 7.4) was pre-heated in water bath at 37 °C for 10 min. The KPF@silica nanofibers were then added to the PBS at 37 °C with stirring. A total of 100 µL PBS was withdrawn at different time intervals and then analyzed by UV-Vis analysis. After analysis, the PBS solution was placed back into the original release medium in order to mitigate the effect of sampling.

### 2.7. Characterization

The morphology of organic and silica nanofibers was examined by scanning electron microscopy (Hitachi S4800 SEM). Silica nanofibers were also imaged using a JEOL 2100+ transmission electron microscope (TEM) operating at 200 KV. Fourier transform infrared (FTIR) spectra were collected using a Vertex 70 FTIR Spectrometer. Powder X-ray diffraction (PXRD) patterns were collected on a Bruker-AXS D8 advanced diffractometer with CuK radiation source. N_2_ sorption analysis was carried out using a Micrometric 3-Flex 3500 Gas Sorption Analyzer. The drug loading and PLGA coating on silica nanofibers were determined by means of thermogravimetric analysis (TGA, Netzsch TG 209 F1 Libra). The concentration of KPF in PBF solutions during the release tests was determined by UV-vis spectrophotometry (a UV plate reader, μQuant, Bio-Tek Instruments).

## 3. Results and Discussion

### 3.1. Organic Nanofibers to Silica Nanofibers

Figure 1 describes the preparation of M-BTC nanofibers by the self-assembly process and the subsequent use of these organic nanofibers to prepare silica nanofibers for drug release. The organic nanofibers were formed by combining an aqueous melamine solution and BTC solution in ethanol. It was suggested that the intermolecular hydrogen-bonding interaction between the amino group of melamine and the carboxyl group of BTC resulted in the self-assembly and formation of M-BTC nanofibers [32,36]. Figure 1A shows the network or random packing of fibers in the region of micrometers. At higher magnifications (Figure 1B,C), it is clear that nanofibers with the diameter of around 50–100 nm are bundled together. These nanofibers are not cylindrical but appear to be shaped fibers with edges. Figure 1D shows the PXRD pattern of the M-BTC nanofibers. The sharp peaks indicate that these nanofibers are highly crystalline. This is generally consistent with the observation in a previous study, where the lattice indices of M-BTC nanofibers were confirmed via crystal X-ray diffraction, indicating a triclinic structure [36].

The FTIR spectrum of M-BTC nanofibers shows the N-H stretching approximately at 3380 cm^−1^ and the C=O stretching at 1690 cm^−1^ (Figure 2A). A lack of a distinct peak at ~1400 cm^−1^ (indicative of C-N stretching) suggests that the covalent amide groups have not been formed [38]. This is also evidenced by the poor stability of these nanofibers in acidic and basic aqueous solutions. The M-BTC nanofibers were thus used as templates to prepare silica nanofibers, in order to improve fiber stability and porosity for drug loading and release.

To produce silica nanofibers, a silica sol was firstly prepared. The M-BTC nanofibers were added and homogenized to form a stable nanofiber suspension which was then frozen in liquid nitrogen and freeze dried. To ensure a silica coating on the nanofibers instead of large pieces of silica between the fibers, the silica sol with low TEOS concentration was used. The sol–gel process at room temperature was very slow. Therefore, after freeze-drying, the materials were heated in an oven at 100 °C for 24 h to complete the sol–gel process and solidify the materials. During this preparation, CTAB of different concentrations were included in the silica sol to induce surfactant templating in the silica materials (Figure 1). Both the M-BTC fibers and CTAB were removed by calcination in air.

After calcining the composite silica/M-BTC fibers, the FTIR spectrum shows two main peaks at 1080 cm^−1^ and 900 cm^−1^, indicating the presence of Si-O-Si vibrations and Si-OH stretching, respectively (Figure 2A). For the thermal stability of M-BTC nanofibers, the TGA profile in Figure 2B reveals that the weight loss reaches nearly 95% when the temperature is increased to around 360 °C. For the silica fibers, as expected, the mass remains at 99.63% when the sample is heated to 600 °C (Figure 2B). The mass loss of 4.76% is mostly attributed to the loss of water adsorbed on silica nanofibers. This suggests that M-BTC nanofibers and CTAB have been successfully removed after the calcination process and that the remaining solid is only silica nanofibers, which is consistent with the result of FTIR analysis.

Figure 3A shows the silica-coated M-BTC fibers prepared with 4 *w/v*% silica sol. The fibrous structure can still be observed, although there are features of silica particles/aggregates. After calcination, a similar fibrous structure is obtained (Figure 3B). This suggests the complete coating of M-BTC fibers. It is understandable that the TEOS concentration in the silica sol can affect the coating. A less concentrated silica sol can, in principle, lead to a thinner coating on the organic fibers. This has been demonstrated by the use of 0.8 *w/v*% silica sol. From Figure 3C, one can see the composite fibers are very similar to the M-BTC fibers. This indicates a thin and uniform coating of silica, without the formation of silica mass between the fibers. This is further demonstrated by the hollow silica nanofibers shown by TEM imaging in Figure 3D. The hollow nanofibers (around 50 nm, consistent with individual M-BTC nanofibers in Figure 1C) aggregate together to form the fibrous morphology as seen in Figure 3C. Silica nanofibers prepared by electrospinning generally show smooth surface morphology, with individual nanofibers separating from each other [20,26,27]. For the nanofibers produced in this work, they are randomly positioned and are observed to be connected by additional silica, which may provide easily accessible porosity and good mechanical stability. For both methods, it is possible to incorporate surfactant templates to tune surface area and mesoporosity in the amorphous silica framework.

### 3.2. N_2_ Sorption Analysis of Silica Nanofibers

As the intended application was to upload drug molecules for controlled release, the surface area and porosity of the silica nanofibers were assessed by N_2_ sorption analysis. The silica nanofibers were prepared without CTAB as a template and with CTAB as a template (CTAB concentrations in the silica sol 1.4 wt%, 2.0 wt%, and 2.6 wt%). Figure 4 shows the sorption isothermals of these silica nanofibers. The hysteresis at the high relative pressure (0.9 P/P_0_ and higher) indicates the presence of macropores, which may have resulted from the voids between nanofibers and the hollow nature of the nanofibers. With the induction of CTAB as a template, a type IV isotherm with an H3 hysteresis begins to appear in the range of 0.45–1.0 P/P_0_ [39]. The H3 hysteresis between adsorption and desorption branches indicates the presence of mesopores (Figure 4).

Figure 5 shows the pore size distribution of the silica nanofibers. The pore size distributions were calculated using the Barrett, Joyner, and Halenda (BJH) method from the desorption data [40]. When CTAB was not included in the silica sol, the silica material only shows the relatively large mesopores peaked at around 15 nm (Figure 5A), which can be mainly attributed to the hollow silica nanofibers (as observed in Figure 3D). No CTAB-templated mesopores around 2 nm are observed. With the inclusion of CTAB in the silica sol, mesopore peaks around 2 nm appear in the silica nanofibers while the large mesopores around 10 nm remain (Figure 5B–D). With the increase of CTAB concentration, the mesopores around 2 nm look sharper (Figure 5C,D). The mesopores around 2 nm are generated from the templates of CTAB micelles. There would be an optimal CTAB concentration for the formation of these mesopores—lower concentration results in lower surface area/pore volume, whilst higher concentration could lead to large aggregates of CTAB.

As shown in Figure 6A, the Brunauer–Emmett–Teller (BET) surface area increases with the increase of CTAB concentration. There is a bigger increase between that without CTAB (concentration = 0) and CTAB concentration 1.4 wt%. The increase in surface area is slowed down with the further increase of CTAB. For the pore volume, it is highest when the CTAB concentration is 1.4 wt% (Figure 6B), a very big jump from CTAB concentration = 0. However, further increase of CTAB concentration leads to a decrease in pore volume. These results demonstrate the effectiveness of CTAB templating (e.g., when CTAB concentration is 1.4 wt%), and the complexity of CTAB templating with increased CTAB concentration and the presence of M-BTC fibers as another template. The larger aggregates of CTAB at higher concentrations and the possible interaction with M-BTC fiber surface could lead to the results shown in Figure 4, Figure 5 and Figure 6.

### 3.3. Drug Loading and Release Behavior from Silica Nanofibers

In order to demonstrate the potential of drug uploading and prolonged release, the silica nanofibers prepared with 1.4 wt% have been selected for further assessment. This is because of their high pore volume and wide pore size distribution, which, in principle, can lead to high drug loading and prolonged constant release.

The model drug KPF was uploaded into silica nanofibers by a solution soaking and solvent evaporation method. KPF loading on silica fibers was calculated via the TGA results. Figure 7A shows that when the temperature reaches around 300 °C, there is no residue left for the pure KPF sample and 5 wt% mass loss for the silica nanofibers. When the silica nanofibers are prepared by soaking in 80 mg/mL and 20 mg/mL KPF solution, the drug loadings are calculated to be 3.8 g/g and 0.7 g/g (based on the mass of silica nanofibers), respectively (Figure 7A). This is a very high drug loading, which can be attributed to the hollow silica nanofibers with mesopores.

The initial drug test showed a fast burst release of KPF in a short period. To address this issue, it was proposed to coat the silica nanofibers with a biodegradable and biocompatible polymer PLGA. To do this, the silica nanofibers were soaked in PLGA solution (with control on soaking time and PLGA concentration) and then taken out to allow the solvent to be evaporated, forming a PLGA coating. The mass ratio of the PLGA coating to the silica nanofibers was calculated based on TGA analysis. Figure 7B shows an example of the mass loss of the PLGA-coated silica fibers compared with PLGA and silica nanofibers. There is a mass loss of 40.12% for PLGA-coated silica (prepared by soaking in a 50 mg/mL PLGA solution for 3 min) when the temperature is nearly 420 °C, while there is a near complete mass loss for PLGA at above 420 °C. This demonstrates the successful coating of PLGA on silica nanofibers.

The tests on KPF released from silica nanofibers (KPF loading 0.7 g/g) were carried out in PBF at 37 °C. In order to minimize the release of KPF during the PLGA coating process, KPF was dissolved in the PLGA solutions at the concentration of 0.7 *w*/*v*%. The concentration of KPF in PBS was determined by the UV-vis plate reader. As shown in Figure 8A, the non-coated silica nanofibers display a very high initial burst release profile, with a higher cumulative release percentage (nearly 67%). There is minimal releasing after around 3 h. However, when the silica nanofibers are coated by PLGA (100 mg/mL and 150 mg/mL), the KPF release could last for up to 25 days. A further release is expected but more long-term release tests need to performed to confirm this. It can be seen from Figure 8A that the silica nanofibers coated with more concentrated PLGA solutions lead to a lower release rate and a lower cumulative release in a given period.

It was found that the PLGA coating time could also affect the release behavior of KPF the from coated silica nanofibers. Figure 8B shows that when the coating time increases, the KPF release is slow and the cumulative release is lower. This effect is more significant when the soaking time is increased from 1 min to 3 min. Further increases in soaking time have a much smaller impact (Figure 8B). This observation may be attributed to a thick PLGA coating formed from a longer soaking time. It is likely that, when the equilibrium between PLGA solution and the soaked silica nanofibers is reached, further increase of the soaking time would not enhance PLGA coating on silica nanofibers.

The results from Figure 8 suggest that PLGA coating on silica nanofibers is an effective way of regulating the KPF release. By varying PLGA concentration and/or soaking time during the coating process, the burst release of KPF at the initial stage can be reduced. It is also possible to achieve the prolonged release of the model drug. Considering that the hollow nanofibers can be used as a reservoir with high drug loading, a long-term release with the desired release profile may be realized by optimizing the porosity of silica nanofibers and the coating/formulation of PLGA and other biocompatible polymers.

Overall, with the use of these hollow silica nanofibers, a very high drug loading may be easily achieved. A coating of functional biopolymers on the silica nanofibers can considerably reduce the undesirable burst release. By varying the type and molecular weight of the biopolymer and the thickness/porosity of the coating, it is viable to realize a prolonged and controlled drug release system. Furthermore, these silica nanofibers can be processed to form three-dimensional networks to act as a scaffold and drug reservoir for potential tissue engineering applications [21,22,23].

## 4. Conclusions

In this study, we developed a new scalable templating method to prepare silica nanofibers with a hollow internal structure. Our method utilized the organic melamine-BTC nanofibers as templates which were formed by a simple self-assembly process under mild conditions. This preparation was solution based and could be readily scaled up. The organic nanofibers were dispersed in silica sol by homogenization. A freeze-drying and sol–gel process was subsequently employed to form silica coating on the melamine-BTC nanofibers. A calcination process was then utilized to generate hollow silica nanofibers. CTAB of different concentrations was dissolved in the silica sol in order to improve the surface area and mesoporosity of the silica nanofiber. These silica nanofibers were shown to be effective scaffolds for drug release, where a model drug KPF was uploaded with a high loading of 3.8 g/g. Importantly, a PLGA coating was formed on the KPF-loaded silica nanofibers to achieve a smaller burst release and prolonged cumulative release. The hollow silica nanofibers with mesoporosity make it easier to achieve very high drug loading and sustainable release. Forming a polymer coating adds another layer of control for prolonging drug release. This method may be extended by uploading different types of drugs, forming coatings with other desirable biopolymer and processing to produce 3D scaffolds for advanced biomedical applications.

## Data Availability

The data presented in this study are available on request from the corresponding author.

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
