# Peer review of "Preparation of PLGA-Coated Porous Silica Nanofibers for Drug Release"

_pharmaceutics, 2022, doi:10.3390/pharmaceutics14122660_

Round 1

Reviewer 1 Report

In this manuscript, the authors reported a new method to prepare silica nanofibers with a hollow internal structure using organic melamine-BTC nanofibers as templates. It was studied the uploading and release of a model drug, ketoprofen.

The characterization techniques have been chosen appropriately for the problem raised. Generally, the results are well discussed providing consistent explanations.

 Some aspects must be clarified before acceptance:

-In the Introduction section the authors should highlight the significance and advantages of their work;

-Why the self-assembly approach can be an alternative to the traditional electrospinning method to prepare silica nanofibers for drug delivery? The authors should make a comparative discussion of the structural, textural, and morphological characteristics of the silica nanofibers obtained by this method with those reported in the literature for the electrospinning  method.

-What improvements does this method bring in terms of drug delivery?

-The conclusion section should be more elaborated.

Reviewer 2 Report

The present work is devoted to the development of an efficient method for the development of hollow silica nanofibers suitable for long-term release with a high drug content. The structure of the article is well organized, and the presented results may potentially be of interest to researchers in the field of drug delivery systems.

Thus, I recommended the manuscript for publication after minor revisions.

The main criticism concerns the introduction. Considerable attention is paid to self-assembly as an alternative method for obtaining nanofibers compare with the electrospinning method. However, the general situation with the use of porous silica materials in the drug delivery system is poorly documented, and new articles are not mentioned.

In addition, the choice of reagents used as a building block for the construction of nanofibers is not explained.

I suggest rewriting the introduction to improve readability and consistency.

Also, fig. 1D needs more explanation regarding the structure of the material, in the text and in the figure.
